# Strong protective effect of the *APOL1* p.N264K variant against G2-associated focal segmental glomerulosclerosis and kidney disease

African Americans have a significantly higher risk of developing chronic kidney disease, especially focal segmental glomerulosclerosis -, than European Americans. Two coding variants (G1 and G2) in the *APOL1* gene play a major role in this disparity. While 13% of African Americans carry the high-risk recessive genotypes, only a fraction of these individuals develops FSGS or kidney failure, indicating the involvement of additional disease modifiers. Here, we show that the presence of the *APOL1* p.N264K missense variant, when co-inherited with the G2 *APOL1* risk allele, substantially reduces the penetrance of the G1G2 and G2G2 high-risk genotypes by rendering these genotypes low-risk. These results align with prior functional evidence showing that the p.N264K variant reduces the toxicity of the *APOL1* high-risk alleles. These findings have important implications for our understanding of the mechanisms of *APOL1*-associated nephropathy, as well as for the clinical management of individuals with high-risk genotypes that include the G2 allele.

African Americans develop kidney disease at a rate five times higher than European Americans[1]. Two African ancestry-associated variants (G1 and G2) in the apolipoprotein L1 (*APOL1*) gene constitute major contributors to this disparity. *APOL1* is a component of the innate immune system targeting African trypanosomes, and the G1 and G2 variants likely rose to high population frequency by conferring resistance to *Trypanosoma brucei rhodesiense* (particularly G2) and *Trypanosoma brucei gambiense* (exclusively G1)[2,3]. However, the putative evolutionary benefits come at a cost of increased lifetime risk for kidney disease in individuals with two copies of these variants (i.e., G1/G1, G2/G2, or G1/G2, identified as *APOL1* high-risk genotypes). This is thought to be mediated by the ability of G1 and G2 variants to form cation-selective channels in podocytes resulting in subsequent activation of cytotoxic pathways[4–6]. This predisposes to progressive kidney disease, with odds ratios for hypertension-associated end stage kidney disease (ESKD), focal segmental glomerulosclerosis (FSGS), and HIV-associated nephropathy exceeding 7, 17, and 30, respectively, when comparing *APOL1* high-risk (APOL1-HR, i.e., individuals carrying either the G1/G1, G1/G2, or G2/G2 genotypes) to low-risk (APOL1-LR) genotypes[3,7].

The number of at-risk individuals for *APOL1*-associated FSGS and kidney disease is considerable. In the United States, it is estimated that 13% of African Americans carry two high-risk alleles[8], and in certain West African populations, the rate of high-risk genotypes may be as high as 20–25%[8,9]. Approximately 15% of individuals with an APOL1-HR genotype will develop ESKD, and a smaller fraction, estimated at 5%–8%, will develop FSGS[8]. Due to the high frequency of these genotypes, we estimate that at least 200,000 individuals in the US have *APOL1*-associated FSGS. The incomplete penetrance of APOL1-HR genotypes is thought to reflect the requirement for disease modifiers that potentiate APOL1 cytotoxicity. A number of "second hits" have been proposed, with the most commonly recognized being high-interferon states (which result in increased APOL1 expression), either due to direct interferon administration, or caused by viral infections (e.g., HIV, SARS-CoV-2)[10,11]. Genetic modifiers have been suggested but, to date, the identification of modifier genetic

e-mail: ss2517@cumc.columbia.edu

variants for *APOL1*-mediated kidney disease and, particularly, FSGS, remains elusive. The few reported in the literature still require validation[12,13].

In 2019, we studied the cytotoxic effect of multiple naturally and non-naturally occurring *APOL1* haplotypes in experimental cell-based systems. We found that the toxicity of G1 and G2 alleles was substantially reduced when expressed on the haplotype defined by the *APOL1* missense variant p.N264K (chr22:36265628 C > A; rs73885316)[14], also associated with a partial loss of trypanolytic function[15]. These data suggested, at a functional level, a protective effect for this variant against the deleterious cellular effects of the G1 and G2 *APOL1* risk variants. The p.N264K defines one of the common G0 (non-risk) APOL1 haplotypes, which is more frequent in individuals of European ancestry, but it is also present on a small fraction of G2 haplotypes in absence of G0, indicating two independent mutational events during evolution only on these two haplotypes. The p.N264K is therefore expected to be mutually exclusive with the APOL1 G1 allele.

In this work we show a strong protective role of the *APOL1* p.N264K variant against APOL1-related FSGS and CKD, in the context of high-risk G2-containing genotypes of African origin. Therefore, this variant, based on prior functional and current genetic data, counters the toxic effect of the G2 allele, allowing the reclassifying of APOL1 high-risk individuals as non-high-risk if they carry the p.N264K missense variant.

## Results

To test the hypothesis that the G2-p.N264K haplotype differs in its genetic impact from the more common G2 risk allele without the p.N264K variant, we sought to compare its frequency in APOL1-HR subjects with FSGS to APOL1-HR controls without kidney disease (Fig. 1A). First, to eliminate potential confounding by the p.N264K haplotype defined by the more common APOL1 non-risk G0 allele, we excluded all individuals with non-risk, G0-containing genotypes, i.e., G0/G0, G0/G1, and G0/G2. We studied two case-control FSGS discovery cohorts: the first consisted of 434 APOL1-HR FSGS cases and 2398 genetically matched APOL1-HR population controls subjected to Illumina DNA microarray genotyping and imputation; the second included 94 APOL1-HR FSGS cases and 208 genetically matched APOL1-HR controls with whole genome sequencing data (Supplementary Fig. 1), for a total of 528 FSGS cases and 2606 population controls with no known kidney disease. Next, in order to investigate the impact of the p.N264K variant, we conducted a comprehensive analysis only on *APOL1* high-risk individuals, employing categorical approaches (based on allelic frequency) and, as sensitivity analysis, regression-based (based on genotypes) statistical tests. The primary analysis was conducted on categorical variables using a Cochran–Mantel–Haenszel (CMH) test and considering potential confounding factors such as sex and array-based vs sequence-based genotyping. We then conducted a set of sensitivity analyses: first, we used Firth's regression test and also incorporated principal

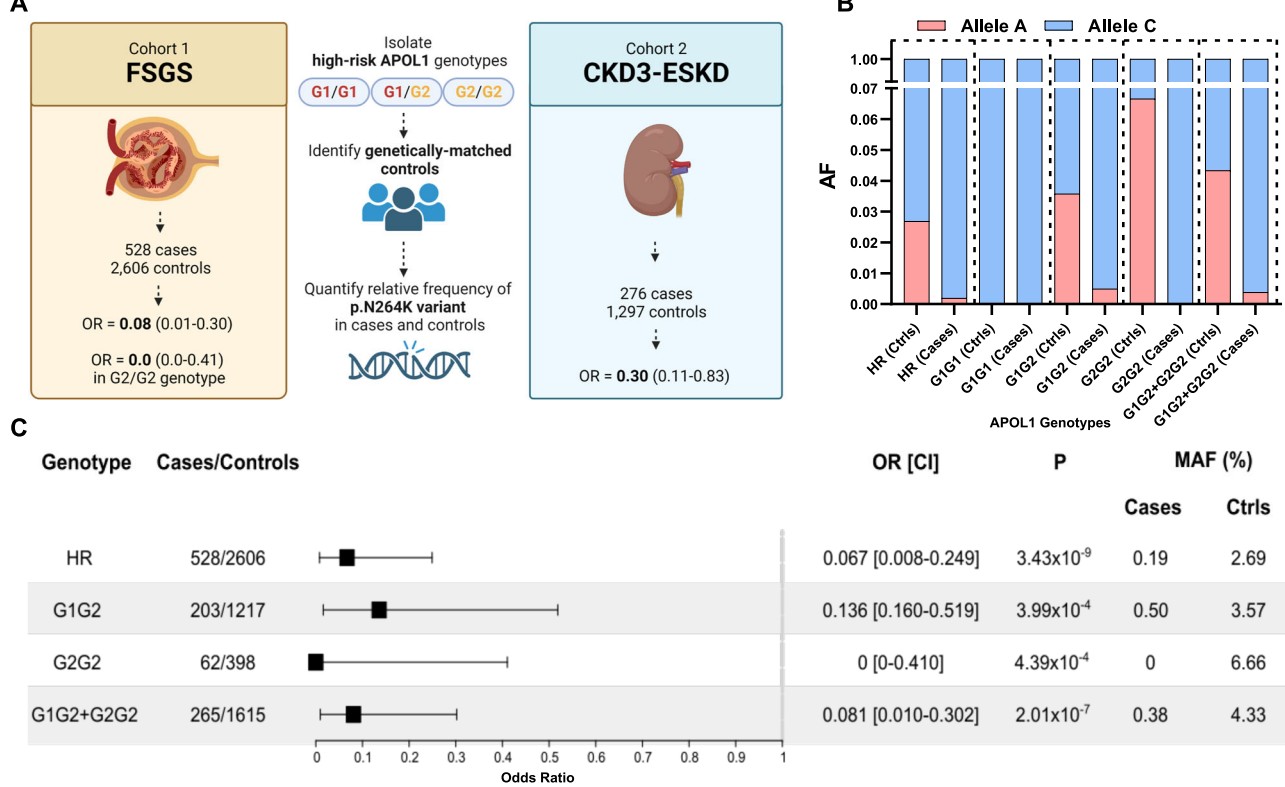

**Fig. 1 | Protective effect of the *APOL1* p.N264K missense variant against G2-associated FSGS. A** Graphical representation of the study design, cohorts and main results of the study. Stratified association analysis of the combined cohort of 528 *APOL1* high-risk FSGS and 2606 genetically-matched *APOL1* high-risk controls: **B** stacked bar plot for the p.N264K MAF across APOL1-HR genotypes in cases and controls; the Allele C is the reference allele encoding for the p.N264; the Allele A is the minor allele resulting the p.K264 variant amino acid; **C** Forest plot for the p.N264K association analysis showing significantly protective odds ratios across APOL1 high risk (G1G1, G2G2 and G1G2) genotypes. The plot describes odd ratio and confidence interval for HR (OR = 0.067), G1G2 (OR = 0.136), G2G2 (OR = 0) and

G1G2 + G2G2 (OR = 0.081). The *P* values were obtained separately for aforementioned individual risk alleles using two sided Cochran-Mantel-Haenszel chi-squared test without multiple correction across the alleles (See Methods). No OR or CI for the G1G1 genotype (263 cases, 991 controls) are shown in forest plot because the p.N264K was absent in both groups (the *APOL1* G1 and p.N264K alleles are mutually exclusive), resulting in undefined OR, infinite CI, and a *p*-value of 1. FSGS focal segmental glomerulosclerosis, CKD chronic kidney disease, ESKD end-stage kidney disease, AF allele frequency, Ctrls controls, OR odds ratio, CI = 95% confidence interval, MAF minor allele frequency. The cartoon in (**A**) has been created using BioRender at www.biorender.com.

components (PCs) as covariates in order to account for potential residual population stratification; second we conducted haplotype-of-origin analysis using Tractor[16].

In our APOL1- HR FSGS cohorts, we observed a strong protective effect for the p.N264K minor allele 'A' (MAF cases = 0.19% and MAF controls = 2.7%, OR = 0.07, 95%CI = 0.01–0.25, CMH test $P = 3.4 \times 10^{-9}$) as compared to APOL1-HR controls. Stratifying the cohort for the three *APOL1* high-risk genotypes showed that this variant was only observed within APOL1-HR individuals carrying the G2 allele (i.e., G1/G2 and G2/G2) and, as expected, never in G1/G1 subjects (Fig. 1, Supplementary Fig. 2). These findings support a protective effect of the p.N264K variant only in the context of G2-containing APOL1-HR genotypes. In fact, the p.N264K variant seemed to confer complete protection against FSGS as it was never observed in cases in the presence of the G2/G2 genotype: OR = 0, 95%CI 0–0.41; CMH test $P = 4.4 \times 10^{-4}$. A strong and significant protective effect was also observed for the G1/G2 genotype with a p.N264K MAF of 3.57% in controls as compared to 0.49% in cases (OR = 0.14, 95%CI:0.16–0.52; CMH test $P = 4.0 \times 10^{-4}$). Consistent with these findings, analyzing individuals with G1/G2 or G2/G2 genotypes combined increased the level of statistical significance for the p.N264K protective effect (OR = 0.08, 95%CI 0.01–0.3, CMH test $P = 2 \times 10^{-7}$).

The FSGS case-control samples were well-matched on principal component analysis (PCA) (Supplementary Fig. 1). In fact, our sensitivity analyses that additionally adjust for population structure confirmed the results obtained by CMH, as Firth's regression tests supported the strong protective effect of the p.N264K variant against FSGS with comparable effect sizes (Supplementary Fig. 2).

As expected from population distribution of haplotypes, in the context of APOL1-HR genotypes, the p.N264K is limited to G2-containing genotypes (i.e., G1/G2 or G2/G2). Nevertheless, a recombination event between the p.N264K and the G1 or G2 alleles (although very unlikely given the proximity of these APOL1 alleles), could result in contamination from the European G0-p.N264K haplotype due to local ancestry admixture. To evaluate this scenario, in our final sensitivity analysis we conducted haplotype-of-origin analysis in the discovery cohort using Tractor[16], a statistical framework that deconvolutes the local haplotypes into ancestral (in this case European and African) haplotypes. This confirmatory analysis showed a significant protective effect of the p.N264K variant exclusively originating from the African haplotype (OR = 0.10, 95%CI = 0.02–0.29, $P = 1.3 \times 10^{-7}$), while the European haplotype was non-significant (OR$_{(ADJ)}$ = 0.74, 95%CI = 0.00–11.37, $P = 0.85$) despite larger sample size (Supplementary Fig. 3). Again, stratifying for G1/G2 or G2/G2 further validated the G2-specific protective effect of the p.N264K variant for the African haplotype (OR = 0.12, 95%CI = 0.02–0.35, $P = 3.53 \times 10^{-6}$) but not for the European haplotype (OR$_{(ADJ)}$ = 0.76, CI = 0.00–12.75, $P = 0.86$).

Overall, these results support a strong protective effect of the *APOL1* p.N264K missense variant against *APOL1*-associated FSGS, but this effect occurs exclusively on G2-containing *APOL1* high-risk genotypes of African origin. In practical terms, based on these analyses, APOL1-HR individuals are at least 8.3 times less likely to develop FSGS if they carry one copy of the p.N264K missense variant.

Finally, to test the generalizability of these findings to milder forms of *APOL1*-associated kidney disease, we investigated the protective effect of the *APOL1* p.N264K in individuals from the REasons for Geographic and Racial Differences in Stroke (REGARDS)[17] and Electronic Medical Records and Genomics Phase III (eMERGE-III)[18] studies. In aggregate, these cohorts included 1573 APOL1-HR individuals with available kidney function data. Of these, 276 had CKD stage 3 (REGARDS, N = 150; eMERGE-III, N = 126) or worse (considered as cases), and 1297 genetically-matched APOL1-HR controls (REGARDS, N = 893; eMERGE-III, N = 404) with estimated glomerular filtration rate (eGFR) > 60 ml/min/1.73 m² (Supplementary Fig. 4A, B). Despite the

smaller sample size, milder form of *APOL1*-associated kidney disease, and incomplete clinical data to classify and exclude unrelated causes for CKD in these cohorts, the findings revealed a direction-consistent protective effect for the p.N264K variant among individuals with the *G2*-APOL1-HR genotypes, by which p.N264K carriers were 3.3 times less likely to have CKD3 or worse (OR = 0.30, 95%CI: 0.11–0.83, CMH $P = 0.023$, Supplementary Table 2 and Supplementary Fig. 4C), with this likely representing an underestimation due to confounders as mentioned above.

## Discussion

Here we report on the strong protective effect of the *APOL1* p.N264K missense variant against G2-mediated FSGS and kidney disease. These findings are also supported by a recent report from the Million Veteran Program, reporting reduced risk for CKD and ESKD in APOL1-HR individuals with this variant[19]. These results have immediate and broad implications for translational research and clinical practice. First, from the genetic standpoint, it is important to note that we observed a very large effect of p.N264K on mitigating the consequences of the G2 risk allele but saw no evidence of this variant on the more common G1 risk allele. As consequence, because p.N264K and G1 alleles are mutually exclusive, this finding raises the possibility of additional genetic modifiers specific to G1 and, in general, identifiable by considering genotype-specific *APOL1* studies. In addition to studies of the *APOL1* high-risk genotype as a single genetic driver, analyses conducted by partitioning cohorts into the three specific *APOL1* high-risk genotypes, although might require larger sample sizes, are likely to provide significant additional insight into the genetics and underlying biology of *APOL1*-associated FSGS and kidney disease. Second, our genetic observations are in agreement with our previous functional studies showing that the p.N264K variant is able to reverse the cytotoxic effect of both G1 and G2 risk variants in cell-based assays[14]. Therefore, conceptually, it may be best to regard the p.N264-G2 and p.K264-G2 simply as different alleles that encode different proteins. As such, they likely adopt different conformations and/or have different activities at the protein level. This will become clearer as we learn more about the APOL1 protein structure(s) in the future.

Taken together, these data support the hypothesis that the p.N264K missense variant negates the toxic effect of the G2 allele, and will allow the reclassification of a fraction of *APOL1* G1/G2 or G2/G2 high-risk individuals as having a non-high-risk genotype if p.N264K is also present. This discovery has substantial, immediate, and clinically-relevant implications. First, individuals affected by CKD or ESKD with *APOL1* G1/G2 or G2/2 high-risk genotypes but with the p.N264K missense variant are unlikely to have APOL1-associated FSGS, and therefore an additional cause (immune, toxic, structural, or others) should be investigated because this will likely result in a different therapeutic approach. Second, importantly, in kidney transplant settings, these results can significantly affect donor selection and both donor kidney, and recipient graft, outcome. In fact, *APOL1* G2-HR donors who are p.N264K positive will likely have kidney outcomes similar to any of the G1G0, G2G0, and G0G0 low-risk donors, thus expanding donors' pool; kidney transplant recipients of a APOL1-HR-p.N264K kidney will likely have low risk for developing de novo FSGS on the graft or graft failure from APOL1-associated kidney disease. Third, incorporation of this knowledge will allow more accurate study design for new intervention trials by which individuals with APOL1-HR-p.N264K genotypes should not be included in the intervention arm as cases since this genotype is genetically and functionally a low-risk genotype. Finally, the knowledge presented here will affect family risk stratification and planning, and, in general, CKD risk ascertainment at the population level.

## Methods

Written informed consent was collected from all participating patients seen at Columbia (and collaborating Institutions) and/or their

guardians in accordance with the Columbia University Institutional Review Board (Protocol AAAC7385) and the policy on bioethics and human biologic samples of AstraZeneca. All internationally recruited patients and/or their guardians were consented according to the Declaration of Helsinki and in compliance with the local ethic committees, as part of the parent IRB protocol approved at Columbia University.

### FSGS cohorts, controls, genotyping and imputation, and association tests

FSGS case-control cohort 1: The cohort consisted of 434 FSGS APOL1-HR cases (Supplementary Table 1) and 2398 APOL1-HR controls. The genotyping of the cases was performed using multiple versions of the Illumina Multi-Ethnic Global Array (MEGA) chips ($n = 196$) that included MEGA 1.0, MEGA 1.1 and MEGA$^{EX}$, and the Illumina HumanOmniExpress-12 ($n = 238$). The controls were genotyped on MEGA1.0 and were selected based on genetic ancestry and *APOL1* genotype status from over 50,000 individuals from the PAGE consortium[20]. We extracted G1 (rs73885319) and G2 (rs71785313) from the MEGA arrays to define *APOL1* high-risk cohort. Also, the p.N264K (chr22:36265628 C > A; rs73885316) variant was included on the MEGA arrays and hence directly genotyped, while imputed with R2 > 0.8 in the Illumina HumanOmniExpress-12[21]. The differences between the chips were corrected first by mapping all the SNPs to a common cluster file in Genome Studio software for individual platforms and further using Snpflip (https://github.com/biocore-ntnu/snpflip) software. In total, we used 767,100 SNPs as input for imputation after quality control, which included filtration for MAF > 1%, missing SNPs <95%, HWE (controls) $P < 0.00001$, and the McCarthy Group Tools (https://www.well.ox.ac.uk/~wrayner/tools/) for strand bias and removal of SNPs that deviated from expected allele frequency using the 1000 Genome Project. The same quality control was applied for cases from HumanOmniExpress-12 separately.

We performed imputation on APOL1-HR cases (MEGA and HumanOmniExpress) and controls (MEGA) together using the TopMed reference imputation panel[22]. SNPs with R2 > 0.8, MAF > 1%, missing SNPs <95%, and HWE (controls) $P > 0.00001$ were retained. All analyses were done on unrelated samples after removing the relatedness up to two degree using KING v2.3.0[23]. PCs were calculated using PLINK 2 based on the LD-pruned SNPs[24].

FSGS case-control cohort 2: The cohort for this study consisted of 94 APOL1-HR cases (refer to Supplementary Table 1) and 208 APOL1-HR controls, all subjected to 30X whole-genome sequencing. To obtain the genetic data, the raw FASTQ files for the cases were aligned to the hg19 assembly[25]. The alignment data underwent processing using the DRAGEN pipeline, resulting in recalibrated GVCFs (Genomic VCFs)[26]. The GVCFs were jointly called using GATK 4.3 separately for samples from the DUKE and CureGN cohorts[27]. For the control group, we included APOL1-HR samples obtained from the 1000 Genomes Project, MESA cohort, and internal controls from the Columbia University Institute for Genomic Medicine[28,29]. Genotypes were extracted after performing internal harmonization. Initially, all the case samples were lifted from the hg19 to the hg38 assembly using the rtracklayer R package[30]. Then, the genotypes from the cases and controls were merged based on common SNPs.

The *APOL1* G1, G2, and p.N264K variants were directly sequenced to obtain specific genetic information. To ensure the analysis was performed on unrelated individuals, we removed relatedness up to two degrees using the KING v2.3.0 software. The same quality control measures were applied to the FSGS cohort 2, as for the initial FSGS cohort, before calculating PCs.

### Statistical analyses

Stratified analysis on alleles was conducted on the two FSGS cohorts using the CMH test statistic. The CMH test was performed using the mantelhaen.test function in R with the exact = TRUE option[31]. The cohorts were stratified based on the variables of cohort and sex for each of the haplotypes (G1/G1, G1/G2, G2/G2) both individually and combined.

Furthermore, Firth regression was performed separately for each haplotype within each cohort using PLINK2[32]. The covariates used in the regression analysis were sex and the first two PCs. Finally, a meta-analysis was conducted to combine the results from the two cohorts. The fixed-effect model was utilized, considering the effect sizes and standard errors obtained from the PLINK2 analysis[33].

### Ancestry resolution analysis at the *APOL1* locus on FSGS cohort 1

Phased genotypes from the first FSGS cohort were utilized after imputation with the TOPMed reference panel. These genotypes were employed for predicting local ancestry inference (LAI) using RFMix v2[34]. The LAI prediction was performed against samples harboring common variants obtained from the 1000 Genomes Project, specifically YRI (representing the African population) and CEU (representing the European population). The output from RFMix was subsequently integrated into the Tractor pipeline to deconvolute ancestry-specific dosages, variant call files), and haplotype counts for each sample and SNP[16]. Within Tractor, ancestry-specific haplotype counts and dosages for the p.N264K variant were extracted. Firth regression analysis was conducted using the logistf R package, incorporating sex, admixture fraction (derived from RFMix), and haplotype counts as covariates in a fixed effect model[32]. This analytical approach facilitated the assessment of the p.N264K variant's association with FSGS by incorporating LAI, deconvolution of dosages and haplotype counts, and regression analysis with appropriate covariates.

### CKD cohorts, genotyping, and analyses

The REGARDS study: The REGARDS study investigates the incidence of stroke in a population of 30,239 Black and White adults (≥45 years of age)[17]. Within this study, we identified 8198 Black participants with genotyped *APOL1* risk alleles (G1 & G2) using TaqMan SNP Genotyping Assay[35] and genome-wide genotyping using the MEGA. To increase sample size, we imputed *APOL1* genotypes for an additional 534 subjects using the TOPMed Imputation Server[21]. Using kinship analysis, we identified and removed related samples (up to 2nd degree) between the REGARDS and PAGE consortiums. Finally, we removed all individuals with an APOL1 low-risk genotype, i.e., G0/G0, G0/G1, G0/G2. Our final cohort was composed of 1043 *APOL1* high-risk individuals with genotypes G1/G1 ($n = 417$), G1/G2 ($n = 455$) and G2/G2 ($n = 171$). The p.N264K variant was included and directly genotyped in the MEGA array. To compare the allele frequency of the p.N264K variant, we stratified samples into the two following groups: cases - estimated glomerular filtration rate (eGFR) < 60, or ESKD, or self-reported kidney failure ($N = 150$); control: eGFR > 60 ($N = 893$). eGFR was measured from the CKD-Epi equation.

Electronic Medical Records and Genomics (eMERGE) study: The eMERGE network has made available electronic health record information connected to GWAS data for a total of 102,138 individuals[18]. These individuals were recruited in three phases (eMERGE-III) from 12 participating medical centers spanning the years 2007–2019. The study cohort consisted of 54% females, with an average age of 69 years. Self-reported demographics indicated that 76% identified as European, 15% as African American, 6% as Latinx, and 1% as East or Southeast Asian.

Every individual underwent genome-wide genotyping, and the specific procedures for genotyping, quality control analyses, and imputation have been previously documented[36]. PCs were derived using FlashPCA[37] on a collection of 48,509 common variants (minor allele frequency ≥0.01) that were independent (pruned in PLINK using the –indep-pairwise 500 50 0.05 command). Imputation of the *APOL1* variants G1 (R2 = 0.998), G2 (R2 = 0.995), and N264K (R2 = 0.8445) was

performed using the TOPMed imputation server, as outlined in the referenced publication[22]. From this cohort, we selected 530 APOL1-HR individuals with available kidney function data in order to classify 126 cases based on eGFR <60 (CKD3-5 or ESKD) and 404 controls (eGFR >60), in the same way as for the REGARDS study above.

## Reporting summary

Further information on research design is available in the Nature Portfolio Reporting Summary linked to this article.

## Data availability

All data supporting the findings described in this manuscript are available in the article and in the Supplementary Information and from the corresponding author upon request. Genome-sequencing data from the CureGN (Accession phs002480.v3.p3) eMERGE network phase III (Accession: phs001584.v2.p2), REGARDS (Accession: phs002719.v1.p1), Population Architecture using Genomics and Epidemiology (PAGE) (Accession: phs000356.v2.p1), and MESA (Accession phs001416.v3.p1) studies is deposited in dbGaP. Illumina DNA microarray data generated for the FSGS cohort 1 analysis was used to extract genetic information for the p.N264K single variant analysis, associated only to a sparse markers map in order to generate principal components for ancestry adjustments. As such, a complete genome-wide analysis of these data has not yet been conducted and the full data will be deposited in dbGaP at completion of the genome-wide analyses. The deidentified individual-level DNA microarray data are currently available to investigators upon request by contacting the corresponding author, Dr. Sanna-Cherchi, at ss2517@cumc.columbia.edu. The corresponding author will provide an initial response within one week from request, and the data will be shared upon establishment of a Data Use Agreement (DUA).

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

## Acknowledgements

We thank the patients and their family members for participating in this study. This research was supported by the Department of Defense (W81XWH-16-1-0451, W81XWH-22-1-0966) and by the National Center for Advancing Translational Sciences, National Institutes of Health (Grant Number UL1TR001873), to S.S.-C., by the National Institute of Health Grant RC2-DK122397, to S.S.-C, M.R.P., F.H. and M.G.S, by the National Institute of Health Grant R01-DK007092 to M.R.P. and D.J.F., and by the National Institute of Health Grant RC2-DK116690 to K.K. and M.K. M.G.S is also supported by R01-DK119380 and the Pura Vida Kidney Foundation. J.B.K. is supported by the Intramural Research Program, NIDDK, NIH. A.G.G. is supported by NIDDK 1U01-DK100876 and DOD W81XWH2110550. K.K. is additionally supported by R01-DK105124, R01-DK136765, R01-LM013061, 2U01-HG008680, U01-AI152960, and 5UL1-TR001873. R.G. is supported by 1R01-DK134347-01, NIH/NICHD 1R21-HD104176-01. F.H. is supported by R01-DK076683. The eMERGE Network was initiated and funded by NHGRI through the following grants: Phase IV: U01HG011172 (Cincinnati Children's Hospital Medical Center); U01HG011175 (Children's Hospital of Philadelphia); U01HG008680 (Columbia University); U01HG011176 (Icahn School of Medicine at Mount Sinai); U01HG008685 (Mass General Brigham); U01HG006379 (Mayo Clinic); U01HG011169 (Northwestern University); U01HG011167 (University of Alabama at Birmingham); U01HG008657 (University of Washington); U01HG011181 (Vanderbilt University Medical Center); U01HG011166 (Vanderbilt University Medical Center serving as the Coordinating Center); Phase III: U01HG8657 (Group Health Cooperative/University of Washington); U01HG8685 (Brigham and Women's Hospital); U01HG8672 (Vanderbilt University Medical Center); U01HG8666 (Cincinnati Children's Hospital Medical Center); U01HG6379 (Mayo Clinic); U01HG8679 (Geisinger Clinic); U01HG8680 (Columbia University Health Sciences); U01HG8684 (Children's Hospital of Philadelphia); U01HG8673 (Northwestern University); U01HG8701 (Vanderbilt University Medical Center serving as the Coordinating Center); U01HG8676 (Partners Healthcare/Broad Institute); and U01HG8664 (Baylor College of Medicine); Phase I–II: U01HG006828 (Cincinnati Children's Hospital Medical Center/Boston Children's Hospital); U01HG006830 (Children's Hospital of Philadelphia); U01HG006389 (Essentia Institute of Rural Health, Marshfield Clinic Research Foundation and Pennsylvania State University); U01-HG006382 (Geisinger Clinic); U01-HG006375 (Group Health Cooperative/University of Washington); U01-HG006379 (Mayo Clinic); U01-HG006380 (Icahn School of Medicine at Mount Sinai); U01-HG006388 (Northwestern University); U01-HG006378 (Vanderbilt University Medical Center); U01-HG006385 (Vanderbilt University Medical Center serving as the Coordinating Center); U01HG004438 (CIDR) and U01HG004424 (the Broad Institute) were serving as phase I Genotyping Centers. The CureGN Study is supported by the National Institute of Health grants 2U01-DK100876. The Nephrotic Syndrome Study Network (NEPTUNE) is part of the Rare Diseases Clinical Research Network (RDCRN), which is funded by the National Institutes of Health (NIH) and led by the National Center for Advancing Translational Sciences (NCATS) through its Division of Rare Diseases Research Innovation (DRDRI). NEPTUNE is funded under grant number U54DK083912 as a collaboration between NCATS and the National Institute of Diabetes and Digestive and Kidney Diseases (NIDDK). Additional funding and/or programmatic support is provided by the University of Michigan, NephCure Kidney International, Alport Syndrome Foundation, and the Halpin Foundation. RDCRN consortia are supported by the RDCRN Data Management and Coordinating Center (DMCC), funded by NCATS and the National Institute of Neurological Disorders and Stroke (NINDS) under U2CTR002818. The REGARDS study is supported by cooperative agreement U01 NS041588 co-funded by the National Institute of Neurological Disorders and Stroke (NINDS) and the National Institute on Aging (NIA), National Institutes of Health, Department of Health and Human Service. Representatives of the NINDS were involved in the review of the manuscript but were not directly involved in the collection, management, analysis or interpretation of the data. The authors thank the other investigators, the staff, and the participants of the REGARDS study for their valuable contributions. A full list of participating REGARDS investigators and institutions can be found at: https://www.uab.edu/soph/regardsstudy/. N.A.L. is supported by the NIH Grant U01-HG011167. Y.G. is supported by the NEPTUNE Fellowship UMINCH-SUBK00018902. A.M. received support from the American Society of Nephrology KidneyCure Ben J. Lipps Research Fellowship. A.M., L.G, W.M., and G.M. are members of the European Reference Network for Rare Kidney Diseases (ERKNet). A.K. is supported by the NIDDK grant 5K25-DK128563-03. The generation of the whole-genome sequencing data in the CureGN Study was supported by AstraZeneca. The content is solely the responsibility of the authors and does not necessarily represent the official views of the National Institutes of Health. L.L. is supported by the NIDDK grant 1K01DK137031-01.

## Author contributions

Conceptualization: S.S.C., M.R.P. M.G.S., D.J.F.; Formal analysis: Y.G., M.T.M., A.K., C.W., J.K., T.Y.L., D.L.B., M.V., E.F., L.L.; Funding acquisition: S.S.C., M.R.P. M.G.S, K.K., A.G.G, R.G., M.K., F.H.; Investigation: S.S.C, M.R.P., D.J.F., A.L.K., G.B.A, M.C.S, A.B., R.J.F.L., E.E.K., C.A.W., J.B.K., W.K.C., M.B., N.S., D.J.C, M.M., J.M., Q.L., B.L., G.J., K.H., A.M., R.W., R.J.C., W.M., P.C., J.R., A.E., Y. L., W.-Q.W, Q.F., C.W., Y.F., I.J.K., M.N., M.R., G.K.D., L.A.L., E.M.L., C.A.W., I.B., S.V.B.P., E.A.O., A.C.S.S., I.P., F.L., L.G., A.A., G.M.G., R.M.,G.M.; Methodology: S.S.C, M.R.P., M.G.S., Y.G., M.T.M., J.M., Q.L., K.K., R.G., D.S.P., S.P., S.M.; Project administration: S.S.C.; Resources: S.S.C., M.R.P., M.G.S, K.K., A.G.G., M.K., R.G., F.H.; Supervision: S.S.C, M.R.P., M.G.S, D.J.F, K.K.; Validation: V.D.D., J.M., Q.L., D.S., M.C., S.E.J, O.M.G, N.S.C., H.T., M.R.I., N.L.; Visualization: Y.G., J.K., B.W., M.T.M.; Writing-original draft: S.S.C, Y.G., M.R.P., M.G.S., D.J.F., R.G.; Writing-review & editing: S.S.C., M.R.P. M.G.S., K.K., Y.G., J.M., M.T.M., A.G.G, D.B.G., M.K., D.J.F., F.H.

## Competing interests

M.R.P. and D.J.F. report research support from Vertex. E.E.K. has received personal fees from Regeneron Pharmaceuticals, 23&Me, Allelica, and Illumina; has received research support from Allelica; and serves on the advisory boards for Encompass Biosciences, Overtone, and Galatea Bio Inc. D.S.P. and S.P. are current employees and stockholders of AstraZeneca. W.K.C. is on the Board of Directors of Prime Medicine and RallyBio. D.B.G. is the Co-founder and CEO of Actio Biosciences. A.G.G. receives a research grant from Natera and has served on advisory boards for Natera through a service agreement with Columbia University. A.G.G. has served on advisory boards for Actio Biosciences, Novartis, Travere, and Alnylam and has stock options for Actio Biosciences. The remaining authors declare no competing interests.

## Additional information

Yask Gupta[1,2], David J. Friedman[3,4], Michelle T. McNulty[5,6], Atlas Khan[1], Brandon Lane[7], Chen Wang[1], Juntao Ke[1], Gina Jin[1], Benjamin Wooden[1], Andrea L. Knob[3,4], Tze Y. Lim[1,8], Gerald B. Appel[1], Kinsie Huggins[7], Lili Liu[1], Adele Mitrotti[9], Megan C. Stangl[7], Andrew Bomback[1], Rik Westland[10], Monica Bodria[11,12], Maddalena Marasa[1], Ning Shang[1], David J. Cohen[1], Russell J. Crew[1], William Morello[13], Pietro Canetta[1], Jai Radhakrishnan[1], Jeremiah Martino[1], Qingxue Liu[1], Wendy K. Chung[14], Angelica Espinoza[15], Yuan Luo[15], Wei-Qi Wei[16], Qiping Feng[16], Chunhua Weng[17], Yilu Fang[17], Iftikhar J. Kullo[18], Mohammadreza Naderian[18], Nita Limdi[19], Marguerite R. Irvin[20], Hemant Tiwari[21], Sumit Mohan[1,22], Maya Rao[1], Geoffrey K. Dube[1], Ninad S. Chaudhary[20], Orlando M. Gutiérrez[20,23], Suzanne E. Judd[21], Mary Cushman[24], Leslie A. Lange[25], Ethan M. Lange[25], Daniel L. Bivona[1], Miguel Verbitsky[1], Cheryl A. Winkler[26], Jeffrey B. Kopp[27], Dominick Santoriello[28], Ibrahim Batal[28], Sérgio Veloso Brant Pinheiro[29], Eduardo Araújo Oliveira[29], Ana Cristina Simoes e Silva[29], Isabella Pisani[30], Enrico Fiaccadori[30], Fangming Lin[31], Loreto Gesualdo[9], Antonio Amoroso[32], Gian Marco Ghiggeri[11,12], Vivette D. D'Agati[28], Riccardo Magistroni[33], Eimear E. Kenny[34,35,36,37], Ruth J. F. Loos[38,39], Giovanni Montini[13,40], Friedhelm Hildebrandt[4,5], Dirk S. Paul[41], Slavé Petrovski[41], David B. Goldstein[42], Matthias Kretzler[43], Rasheed Gbadegesin[7], Ali G. Gharavi[1], Krzysztof Kiryluk[1], Matthew G. Sampson[4,5,6], Martin R. Pollak[3,4] & Simone Sanna-Cherchi[1] ✉

[1]Department of Medicine, Columbia University Irving Medical Center, New York, NY, USA. [2]Institute for Inflammation Medicine, University of Lubeck, Lübeck, Germany. [3]Nephrology Division, Department of Medicine, Beth Israel Deaconess Medical Center, Boston, MA, USA. [4]Harvard Medical School, Boston, MA, USA. [5]Division of Pediatric Nephrology, Boston Children's Hospital, Boston, MA, USA. [6]Kidney Disease Initiative and Medical and Population Genetics Program, Broad Institute, Boston, MA, USA. [7]Division of Nephrology, Department of Pediatrics, Duke University School of Medicine, Durham, NC, USA. [8]Unit of Genomic Variability and Complex Diseases, Department of Medical Sciences, University of Turin, Turin, Italy. [9]Department of Precision and Regenerative Medicine and Ionian Area (DiMePre-J) Nephrology, Dialysis and Transplantation Unit, University of Bari Aldo Moro, Bari, Italy. [10]Department of Pediatric Nephrology, Emma Children's Hospital, University of Amsterdam, Meibergdreef 9, Amsterdam, The Netherlands. [11]Division of Nephrology and Renal Transplantation, IRCCS Istituto Giannina Gaslini, Genoa, Italy. [12]Laboratory on Molecular Nephrology, IRCCS Istituto Giannina Gaslini, Genoa, Italy. [13]Pediatric Nephrology, Dialysis and Transplant Unit, Fondazione IRCCS Ca' Granda-Ospedale Maggiore Policlinico, Milano, Italy. [14]Departments of Pediatrics and Medicine, Columbia University Irving Medical Center, New York, NY, USA. [15]Center for Genetic Medicine, Feinberg School of Medicine, Northwestern University, Chicago, IL, USA. [16]Division of Clinical Pharmacology, Department of Medicine, Vanderbilt University Medical Center, Nashville, TN, USA. [17]Department of Biomedical Informatics, Columbia University Irving Medical Center, New York, NY, USA. [18]Atherosclerosis and Lipid Genomics Laboratory, Mayo Clinic, Rochester, MN, USA. [19]Department of Neurology, Heersink School of Medicine, University of Alabama at Birmingham, Birmingham, AL, USA. [20]Department of Epidemiology, School of Public Health, University of Alabama at Birmingham, Birmingham, AL, USA. [21]Department of Biostatistics, School of Public Health, University of Alabama at Birmingham, Birmingham, AL, USA. [22]Department of Epidemiology, Mailman School of Public Health, Columbia University, New York, NY, USA. [23]Division of Nephrology, Department of Medicine, Heersink School of Medicine, University of Alabama at Birmingham, Birmingham, AL, USA. [24]Department of Medicine and Pathology and Laboratory Medicine, University of Vermont, Burlington, VT, USA. [25]Department of Biomedical Informatics, University of Colorado Anschutz Medical Campus, Aurora, CO, USA. [26]Cancer Innovation Laboratory, Center for Cancer Research, National Cancer Institute, National Institutes of Health and Basic Research Program, Frederick National Laboratory, Frederick, MD, USA. [27]Kidney Disease Section, National Institute of Diabetes and Digestive and Kidney Diseases (NIDDK), NIH, Bethesda, MD, USA. [28]Department of Pathology and Cell Biology, Columbia University Irving Medical Center, New York, NY, USA. [29]Universidade Federal de Minas Gerais (UFMG), Faculdade de Medicina, Laboratório Interdisciplinar de Investigação Médica, Departamento de Pediatria, Unidade de Nefrologia Pediátrica, Belo Horizonte, MG, Brazil. [30]Nephrology Unit, Parma University Hospital, and Department of Medicine and Surgery, University of Parma, Parma, Italy. [31]Division of Pediatric Nephrology, Department of Pediatrics, Columbia University, New York, NY, USA. [32]Immunogenetics and Transplant Biology Service, University Hospital "Città della Salute e della Scienza di Torino", Department of Medical Sciences, University of Turin, Turin, Italy. [33]Surgical, Medical and Dental Department of Morphological Sciences, Section of Nephrology, University of Modena and Reggio Emilia, Modena, Italy. [34]Institute for Genomic Health, Icahn School of Medicine at Mount Sinai, New York, NY, USA. [35]Department of Genetics and Genomic Sciences, Icahn School of Medicine at Mount Sinai, New York, NY, USA. [36]Center for Translational Genomics, Icahn School of Medicine, New York, NY 10027, USA. [37]Division of Genomic Medicine, Department of Medicine, Icahn School of Medicine, New York,

NY 10027, USA. [38]The Charles Bronfman Institute for Personalized Medicine, Icahn School of Medicine at Mount Sinai, New York, NY, USA. [39]Novo Nordisk Foundation Center for Basic Metabolic Research, University of Copenhagen, Copenhagen, Denmark. [40]Department of Clinical Sciences and Community Health, Giuliana and Bernardo Caprotti Chair of Pediatrics, University of Milano, Milano, Italy. [41]Centre for Genomics Research, Discovery Sciences, Bio-Pharmaceuticals R&D, AstraZeneca, Cambridge, UK. [42]Institute for Genomic Medicine, Columbia University Irving Medical Center, New York, NY, USA. [43]Department of Internal Medicine, Division of Nephrology, University of Michigan, Ann Arbor, MI, USA. ✉e-mail: ss2517@cumc.columbia.edu

