## [Peer Review File · Nature Communications]

Strong protective effect of the APOL1 p.N264K variant against G2-associated focal segmental glomerulosclerosis and kidney diseaseREVIEWER COMMENTS

Reviewer #1 (Remarks to the Author):

Sanna-Cherchi and colleagues present analyses on the influence of APOL1 gene variants on focal segmental glomerulosclerosis and chronic kidney disease. Two African ancestry-associated variants in APOL1 gene (G1 and G2) contribute to the five times higher rate of kidney disease in Black Americans compared to European Americans. Here, the authors searched for an explanation why the penetrance of the two APOL1 variants for end-stage kidney disease and focal segmental glomerulosclerosis is incomplete. Other APOL1 variants may act as genetic modifiers. Haplotype-based analysis with G1, G2 and another missense mutation (p.N264K) was the motivation of this work. The G2/p.N264K haplotype was analyzed for differences in genetic impact in comparison to the more common G2 risk allele. G0 containing genotypes were excluded to eliminate potential confounding, since G0/p.N264K and G2/p.N264K are evolutionary independent haplotypes.

The interpretation of the results is reasonable. In general, the data analysis and the statistical methods are state-of-the-art. However, some minor concerns regarding the analyses and presentation of the results should be addressed:

1. In the middle of the text (and in Figure 1) the expression APOL1-HR occurs without explanation. From Figure 1 can be deduced that the APOL1-high risk genotypes are meant (G1G2, G2G2, G1G2 + G2G2), however, the sum of the number is inconclusive. In Figure 1C: Why do the numbers of the groups G1G2 (203/1217) and G2G2 (62/398) not sum to the total number of HR (528/2606)? Is the difference representing the G1G1 group? I guess so, but it would be helpful for the reader to clarify that. From Figure 1B it is obvious, that G1 and p-N264K are mutually exclusive. Very helpful figure!
2. Why are the data from the chronic kidney disease and end-stage kidney disease analysis not presented graphically (forest plot)? At least as a supplementary figure after Suppl. Table S2? In the text, Supplementary Figure S4 is cited but not included in the manuscript (the provided Supplementary Figure S4 is principal component plot).
3. Supplementary Appendix:
 - a. Please cite the McCarthy Group tools for strand bias (line 52).
 - b. The REGARDS study: The numbers of the CKD cases and controls are not stated (250/893, I guess).
 - c. line 125: APOL1 italic
4. Maybe a comparison between the population attributable fraction of APOL1-HR with and without p-N264K would be interesting.

Overall, this is a very interesting manuscript which is clearly written. After addressing the above-mentioned minor concerns, it would be of great interest for researchers in the field of kidney disease, especially because of the clinical implications mentioned at the end of the discussion.

Reviewer #2 (Remarks to the Author):

In this manuscript Gupta et al. demonstrate that APOL1 p.N264K variant acts protectively against G2 associated FSGS and CKD.

N264K variant in the membrane-addressing domain of APOL1 has been initially described in association with increased susceptibility to African Trypanosomiasis due to a loss of trypanolytic function, and has been subsequently shown to reduce the toxicity of G1 and G2 alleles.

In the current communication the authors investigate the protective role of this variant in two large FSGS cohorts having APOL1 HR genotype. They demonstrate that p.N264K indeed has a strong protective role, but only in G2 genotypes. In addition, they evaluated another two cohorts to investigate the protective effect of N264K on CKD3 or more, and, although the effect was less pronounced, it still could be demonstrated.

The paper contains important information for the field and has direct therapeutic consequences. Moreover, it feeds fundamental research towards further investigation of the mechanism of this protective effect. The manuscript is well written and the methodology is solid.

I have two remarks to consider:

1) The authors excluded low risk genotypes to eliminate potential confounding of G0 allele. Although I can follow their argument, I would consider important to demonstrate that in low risk genotypes the protective effect of p.N264K was absent. Has this analysis been performed or can it still be performed?

2) In the discussion section, it will be interesting to read the thoughts of the authors regarding a possible mechanistic explanation of the observed protective effect.

Point-by-point response to Reviewers

First of all, we want to sincerely thank both reviewers for the timely, constructive, and positive comments to our work, which we believe helped in significantly improving the manuscript.

Reviewer #1 (Remarks to the Author):

Sanna-Cherchi and colleagues present analyses on the influence of APOL1 gene variants on focal segmental glomerulosclerosis and chronic kidney disease. Two African ancestry-associated variants in APOL1 gene (G1 and G2) contribute to the five times higher rate of kidneys disease in Black Americans compared to European Americans. Here, the authors searched for an explanation why the penetrance of the two APOL1 variants for end-stage kidney disease and focal segmental glomerulosclerosis is incomplete. Other APOL1 variants may act as genetic modifiers. Haplotype-based analysis with G1, G2 and another missense mutation (p.N264K) was the motivation of this work. The G2/p.N264K haplotype was analyzed for differences in genetic impact in comparison the more common G2 risk allele. G0 containing genotypes were excluded to eliminate potential confounding, since G0/p.N264K and G2/p.N264K are evolutionary independent haplotypes.

The interpretation of the results is reasonable. In general, the data analysis and the statistical methods are state-of-the-art. However, some minor concerns regarding the analyses and presentation of the results should be addressed:

RESPONSE:

Thank you for the positive evaluation of our work

1. In the middle of the text (and in Figure 1) the expression APOL1-HR occurs without explanation. From Figure 1 can be deduced that the APOL1-high risk genotypes are meant (G1G2, G2G2, G1G2 + G2G2), however, the sum of the number is inconclusive. In Figure 1C: Why do the numbers of the groups G1G2 (203/1217) and G2G2 (62/398) not sum to the total number of HR (528/2606)? Is the difference representing the G1G1 group? I guess so, but it would be helpful for the reader to clarify that. From Figure 1B it is obvious, that G1 and p-N264K are mutually exclusive. Very helpful figure!

RESPONSE:

We thank the reviewers for pointing out this. According to the suggestion we have clarified the *APOL1-HR* genotypes and their number in the manuscript. The Reviewer is right about the G1G1 and to clarify Figure 1C, we have added the number of individual with G1/G1 genotype in the legend.

2. Why are the data from the chronic kidney disease and end-stage kidney disease analysis is not presented graphically (forest plot)? At least as a supplementary figure after Suppl. Table S2? In the text, Supplementary Figure S4 is cited but not included in the manuscript (the provided Supplementary Figure S4 is principal component plot).

RESPONSE:

We thank the reviewer for the comment. We have added the forest plot for REGARDS and eMERGE-III as supplementary Figure 4C.

3. Supplementary Appendix:

a. Please cite the McCarthy Group tools for strand bias (line 52).

b. The REGARDS study: The numbers of the CKD cases and controls are not stated (250/893, I guess).

c. line 125: APOL1 italic

RESPONSE:

We appreciate the reviewer for pointing out these minor details for good scientific practice. We have added the citation for McCarthy Group tools, added the number of cases and controls with REGARDS study and made *APOL1* italic.

4. Maybe a comparison between the population-attributable fraction of APOL1-HR with and without p-N264K would be interesting.

RESPONSE:

We concur with the reviewer that a comparative analysis would yield significant insights. Based on our data and available epidemiological data we could estimate the following: The United States is home to a population of 49,586,352 African-American individuals (source: <https://www.census.gov/newsroom/facts-for-features/2023/black-history-month.html>). Approximately 13% of African-Americans carry *APOL1-high risk* genotypes, leading to about 6,446,225 individuals. In our 2,606 *APOL1-HR* population controls, 61.97% were either G1/G2 (46.70%) or G2/G2 (15.27%). Extrapolating from our data, approximately 3,996,659 Black Americans (8.06% of the total) carry a G2-containing *APOL1-HR* genotypes.

In our investigation, we observed robust protective effects of the p.N264K for the G1/G2 and G2/G2 genotypes. We assume that heterozygous and homozygous p.N264K genotypes have equivalent protective effect (dominant model). We identified a total prevalence of 8.4% for p.N264K genotypes among *APOL1-G2*-containing HR genotypes among the 1,491 controls in the FSGS cohort 1. Similarly, in the FSGS cohort 2, we observed a prevalence of 8.87% for p.N264K genotypes among 124 controls. Based on extrapolation from these estimated frequencies, approximately 339,716 African American individuals (calculated as 3,996,659 x 8.5%) could be immediately reclassified from the *APOL1-High Risk (APOL1-HR)* group to the *APOL1-Low Risk (APOL1-LR)* group. This translate in one every 12 individuals with a G2-containing *APOL1-HR* genotype being in fact *APOL1-LR*. This is even more striking for the G2/G2 only genotype. In fact, in this genotype category, the prevalence of the p.N264K variants was 12.6%. Applying our genotype frequency data (calculated as 3,996,659 x 6,446,225 x 15.27% x 12.6%), 124,026 out of 984,338 *APOL1-G2/G2* carriers (one in 8) originally classified as *APOL1-HR*, should can be reclassified as *APOL1-LR*.

Nevertheless, we caution to provide such estimates in a discovery study. If the Editors and Reviewers strongly feel for such data, or on a similar line, to be included into the text, we will be happy to do so, but we feel that this “narrative” would suit better as an Editorial accompanying the manuscript or included in some review/opinion article at this moment. Subsequent replication and population-based studies will help us providing more accurate population-attributable fraction estimates of *APOL1-HR* with and without p.N264K. **Thank you Overall, this is a very interesting manuscript which is clearly written. After addressing the above-mentioned minor concerns, it would be of great interest for researchers in the field of kidney disease, especially because of the clinical implications mentioned at the end of the discussion.**

RESPONSE:

Thank you very much for the enthusiastic comment. The potential and immediate clinical impact of these findings is a motivation for our work.

Reviewer #2 (Remarks to the Author):

In this manuscript Gupta et al. demonstrate that APOL1 p.N264K variant acts protectively against G2 associated FSGS and CKD.

N264K variant in the membrane-addressing domain of APOL1 has been initially described in association with increased susceptibility to African Trypanosomiasis due to a loss of trypanolytic function, and has been subsequently shown to reduce the toxicity of G1 and G2 alleles.

In the current communication the authors investigate the protective role of this variant in two large FSGS cohorts having APOL1 HR genotype. They demonstrate that p.N264K indeed has a strong protective role, but only in G2 genotypes. In addition, they evaluated another two cohorts to investigate the protective effect of N264K on CKD3 or more, and, although the effect was less pronounced, it still could be demonstrated.

The paper contains important information for the field and has direct therapeutic consequences. Moreover, it feeds fundamental research towards further investigation of the mechanism of this protective effect. The manuscript is well written and the methodology is solid.

RESPONSE:

Thank you for the very positive reception of our manuscript. We truly appreciate it.

I have two remarks to consider:

1) The authors excluded low risk genotypes to eliminate potential confounding of G0 allele. Although I can follow their argument, I would consider important to demonstrate that in low risk genotypes the protective effect of p.N264K was absent. Has this analysis been performed or can it still be performed?

RESPONSE:

Thank you, this is an incredibly important issue but not as simple to address. We have conducted, with the available data, some analyses in that regard and we indeed do not detect a protective effect of the p.N264K variant in individuals with *APOL1-LR* genotypes. We do not feel though that our current analyses stand the required burden of proof to make these claims for the following reasons. First, the hypothesis behind the protective effect of the p.N264K variant is predicated on the direct effect on the modification of the APOL1 protein, hence the biological mechanism outside of APOL1-mediated kidney disease remains elusive and object of debate. Second, when moving to individuals with *APOL1-LR* genotypes, we deal with individuals who carry G0-containing genotypes, which, in turn, can independently carry the G0-associated p.N264K variant. This proves challenging in dissecting the combination of different haplotypes (the G0 and G2) both containing the p.N264K variant. In fact, in this scenario, all LR genotypes can carry such variant (G0/G0, G0/G1, G0/G2) but on different haplotypes and hence encoding different proteins: it is conceivably different from a conformational standpoint, to have a APOL1 protein that carries both the G2 variant and the p.N264K, as compared to the case of an individual who carries a wildtype APOL1 protein in trans with an APOL1 carrying the G2-p.N264K protein, or compared to an individual who carries a heterozygous APOL1 G2 protein in trans to a heterozygous p.N264K protein (this would be the case of a G0-p.N264K haplotype with a G2 allele without the p.N264K). And more combinatorial scenarios are possible. A way to disentangle this problem is to conduct genotype- or haplotype- specific GWASs, thus comparing, independently, FSGS cases and controls carrying the exact same APOL1 haplotypes. This will require significantly larger sample size. Third, when moving to *APOL1-LR* individuals, we also enrich for global and local European ancestry, thus introducing more ancestral variability. While we are well suited to adjust for both global and local ancestry, this still needs to be taken into account when trying to extrapolate effect sizes across populations. Fourth, in *APOL1-LR* FSGS, hence in absence of *APOL1* high-risk genotypes as drivers of disease, we increase heterogeneity for disease causation, where Mendelian genetic mutations, affecting up to 10-20% of the FSGS cases, or strong immune factors are at play. On one hand it is difficult to envision a situation in which the p.N264K variant would act in both scenarios. on the other hand, we are also currently blind to Mendelian mutations in most of our patients

reported here, since these were mostly subjected to DNA microarray genotyping. We would need to conduct exome or genome sequencing in all our cases so that we could partition our cohort in Mendelian and non-Mendelian FSGS to then conduct the proper p.N264K association analyses.

We are actively working on aggregating larger cohorts with complete sequencing data to address these issues. Because of all the reasons above, we strongly feel that putting out confounded and preliminary results about the lack of protective effect of the p.N264K variant in *APOL1-LR* individuals would not serve well the community, and would also distract the readers from the more directly applicable message of this manuscript.

2) It the discussion section, it will be interesting to read the thoughts of the authors regarding a possible mechanistic explanation of the observed protective effect.

RESPONSE:

Thank you, we added a generic sentence addressing the possible mechanistic explanation. This of course is a task for future studies that we hope we and other investigators will be able to resolve soon.

REVIEWERS' COMMENTS

Reviewer #1 (Remarks to the Author):

The authors have addressed to all my minor concerns in their revised manuscript. I agree with the authors that providing numbers for population attributable fraction from the discovery study is not the best way. But it was interesting for me to see the numbers. Thank you!

The authors did a very good job.

Reviewer #2 (Remarks to the Author):

No comments